# Analysis of the Displacement of Thin-Walled Workpiece Using a High-Speed Camera during Peripheral Milling of Aluminum Alloys

**DOI:** 10.3390/ma14164771

**Published:** 2021-08-23

**Authors:** Jakub Czyżycki, Paweł Twardowski, Natalia Znojkiewicz

**Affiliations:** Faculty of Mechanical Engineering, Poznan University of Technology, ul. Piotrowo 3, 60-965 Poznan, Poland; pawel.twardowski@put.poznan.pl (P.T.); natalia.w.znojkiewicz@doctorate.put.poznan.pl (N.Z.)

**Keywords:** milling, deformation, thin-walled workpiece, aluminum alloys, high-speed camera

## Abstract

The paper presents the possibilities of a high-speed camera in recording displacements of thin-walled workpiece during milling made of aluminum alloys, which allowed for an analysis in which it was compared to other methods of testing the deflection of such elements. The tests were carried out during peripheral milling with constant cutting parameters. Deflection of thin-walled workpiece due to cutting forces was measured using a high-speed camera and a laser displacement sensor. Additionally, the experimental results were compared with the theoretical results obtained with the use of the finite element method. The research proved the effectiveness of the use of high-speed camera in diagnostics of thin-walled workpieces during milling with an accuracy of up to 11% compared to measurements made with a displacement laser sensor.

## 1. Introduction

Thin-walled constructions are increasingly used in various fields of industry, allowing to obtain monolithic structures that eliminate the need to perform assembly operations from multiple components. The advantages that give the ability to create structures made of thin-walled parts are difficulties in their implementation related to deformation under the influence of cutting forces resulting in deterioration of the surface quality, permanent deformation of the element and difficulties or impossibility to assemble [1,2,3].

The use of HSM (high-speed machining) allows for the simultaneous reduction of the cutting force components, which is very important when machining thin-walled workpieces, minimizing their deformation and at the same time obtaining a good quality of the treated surface. In addition, this treatment on the example of Bałoń et al. tests gives the possibility of reducing the machining time four times compared to conventional machining [2,4].

A number of studies focus on predicting the behavior of thin-walled workpieces during cutting through finite element studies or on the basis of general mathematical models or those developed specifically for a given application [5,6].

The authors of the study [6] focused on predicting thermal loads and deformations of a thin-walled workpiece made of aluminum alloy used in the aviation industry. Laser sensors at two measuring points were used to measure the deformation. The results obtained from the finite element analysis were compared to the experimental test, obtaining the prediction of deformations and thermal load of the tested element with an error of 15–25%.

Other studies, in turn, lead to the correction of the tool path and the selection of optimal technological parameters in order to minimize the deformation of the processed materialand the possibility of chatter [7,8,9,10].

As described earlier, the formation of chatter during the machining of thin-walled workpieces has a very negative impact on the quality of their surface. In order to minimize this risk, the authors [11] focus on the study of the natural frequency of thin walls with their various dimensional variants with the use of an automatic ball launcher. The results obtained using this method do not differ from the traditional method of measurement with the use of a modal hammer, additionally they were compared to analytical methods (including FEM). The biggest advantage of this method is the possibility of its implementation in industry when machining thin-walled workpieces in order to control the possibility of self-excited vibrations formation.

The research [7] made it possible to reduce the flatness deviation of the milled element by 45% by obtaining an optimized tool path by predicting the deflection of a thin-walled workpiece using the finite element method. Similarly, Soori and Asmael [10] presented the results of the tool path optimization based on the developed algorithm based on the finite element method and the analytical model. Compensation allowed them to reduce deformations by half.

One of the methods of reducing the deformation of thin workpieces during their processing is the use of special machining fixtures or supporting them (these methods arenot always possible), nevertheless, they themselves can be the cause of post-machining deformations after removing the clamping forces [1,12,13]. Zha et al. [14] avoided the need to use additional supports and fixtures when milling thin-walled workpieces made of titanium alloys by selecting an appropriate machining strategy. Semi-finishing machining with large depths of cut and after that finishing next step with the previouos layer allowed them to shorten the machining time by 40% while maintaining the required quality of the processed surface.

High-speed cameras are currently used in machining to a small extent, they are mainly used in the diagnostics of machine tools to measure the accuracy of positioning [15], tool displacement [16] and in the field of material strength [17,18,19].

High-speed cameras available on the market that allow for recording in a large number of frames per second, while maintaining the resolution that allows to register changes in the cutting process, cost tens of thousands of dollars, the authors [20] have developed a high-speed camera, the cost of which was a fraction of the amount indicated earlier. The camera allowed to record in 2329 frames per second with resolution 256 × 256. By increasing the possibilities of recording in a greater number of frames per second, it would allow the use of such a camera in the diagnosis of machining processes.

The use of a high-speed camera during the shear test allowed for the precise determination of the three shear stages of the tested titanium alloys [18]. On the other hand, the research [21] allowed to measure temperature changes with the use of a high-speed infrared camera during the perforation of ES mild steel. Kuczmaszewski et al. [22] carried out a measurement of the chip temperature during drilling of a magnesium alloy in order to determine the possible risk of ignition during machining. Among the tested technological parameters, the depth of cut had the greatest impact on the chip temperature.

In the study [23], a high-speed camera was used to analyze the morphology of chip formation in the cutting process, the study was additionally confirmed with an analytical model and an inconsiderable influence of the cutting speed change on the geometric features of the chip. On the other hand, Guo et al. [24] presented a method of using a high-speed camera to analyze the dynamics of flow, deformation fields and chip formation in the process of sliding and cutting H02 bronze. The research allowed for the in situ analysis of the impact angle on the chip form in the range of −70° to +15°. The use of a high-speed camera and a merchant circle diagram allowed for the analysis of cutting force, torque and the amount of material removed during turning. However, the obtained results were not compared to the actual values mentioned by the authors [25].

The authors of Rypin et al. used a high-speed camera to analyze the course of cutting with a single abrasive grain in order to create a geometric model that was used to develop a computer model of micro-cutting with a single abrasive grain [26].

Berezvai, Bachrathy and Stepan successfully used a high-speed camera to detect the formation of chatter during orthogonal cutting. In addition, this method can be effectively used with small diameter cutters, where accelerometric measurements does not meet the possibility of measurement. On the other hand, there is no comparison of the detection of chatter to other methods, for example the vibration acceleration sensor, as a method commonly used in this type of test [27].

The aim of the work is to estimate the possibility of high-speed camera in predicting deflections of thin-walled workpiece, there is no literature work focusing on this subject.

## 2. Materials and Methods

### 2.1. Material and Technological Parameters

Machining of thin walls made of 7075 aluminum alloys was carried out on an AVIA FND 32 tool milling machine (factory of precision machine tools Avia, Warsaw, Poland) with a solid carbide end mill (FRAISA C15589391) with a diameter of *D* = 8 mm, the number of teeth *z* = 6, tool helix angle *λ_s_* = 40° adapted to the finishing of aluminum alloys. The initial wall thickness *t* was 2 mm, the wall height and hence the cutting depth *a_p_* = 10 mm, the cutting width being the thickness of the layer of the cut material during a single pass was *a_e_* = 0.2 mm, the machining was carried out at the cutting speed *v_c_*= 110 m/min for feed per tooth *f_z_* = 0.02 mm/tooth. The technological parameters were selected on the basis of Fraisa company recommendations for processing with the selected tool for the material being processed and the possible parameters to be implemented on the milling machine used.

### 2.2. Thin-Walled Workpiece

Figure 1 shows a schematic of a thin-walled workpiece that was the subject of the study. The initial wall thickness was 2 mm with a height of 10 mm. The thickness of the milled walls was realized in the range of 2–0.5 mm, which at a height of 10 mm gives the ratio of height to thickness corresponding to thin-walled parts used in industry [4,5,7].

### 2.3. Experimental Setup

The diagram below (Figure 2) shows the experimental setup, the Phantom miro m310 high-speed camera (Vision Research, Wayne, NJ, USA) and the Micro-Epsilon optoNCDT ILD1700-10 LL displacement laser sensor (Micro-Epsilon, Ortenburg, Germany) with an accuracy of 0.5 µm were used to test the deflection of thin walls. The wall deflection was analyzed at the moment when the end mill exited the last stage of cutting the assumed cutting layer. Tema Motion computer software allowed to obtain the course of thin wall deflection by analyzing the position of the marked point (wall corner) during machining. Figure 3 shows a photo of the test stand. The high-speed camera recorded the image in 384 × 288 resolution at 16,000 frames per second.

## 3. Results

Figure 4 shows an example of a frame from a film recorded with a high-speed camera while milling a thin wall. The measurement point was marked in the form of the left corner, the position of which was monitored during the recording. The recorded sample waveform is shown in Figure 5. The graph shows the moment when the cutter exited the machined wall.

The following test results show the maximum elastic deflection of a thin wall at its top at the stage of completing a single end mill pass by cutting a given material thickness and reaching up to 180 μm. This value is of great importance because it allows to determine whether the deformation is large enough to reduce the assumed cutting layer (*a_e_*), which will cause an error in the thickness of the wall being made, as proven in the tests. The analyzed recordings and deformations obtained from laser sensor confirm the absence of plastic deformation in the entire thickness range of the milled thin-walled workpiece [28].

The diagram below (Figure 6) shows an example of measuring the deformation of a machined thin wall with the use of a laser displacement sensor. The plot shows the moment when the cutter approaches the measurement point, where the deformation reaches its maximum value, and then the deformation disappears with the moment when the tool loses contact with the machined wall. This confirms the absence of plastic deformation.

Figure 7 shows the dependence of the influence of wall thickness *t* on the deflection during milling, measured simultaneously with a displacement laser sensor and a high-speed camera. Both characteristics weresimilar to each other in the entire range of the thickness of the cut walls. Below 1 mm of wall thickness, a sharp increase in deformation of up to 200 µm at a thickness of t 0.4 mm couldbe observed. Measurement of thin wall deformations allowed for an accuracy of up to 11% compared to the displacement laser sensor.

Additionally, the Tema Motion software enabled the amplitude-frequency analysis of the displacement of the machined wall from the recording made using FFT (Figure 8). There weretwo basic dependencies on it:*f_o_*—frequency related to the spindle speed *n*, *f_o_* = *n*/60,*f_oz_*—frequency of the milling process *f_oz_* = *f_o_* × *z*.

In all analyzed spectra of displacement of milled walls, the greatest amplitudes occurred for the signal component with the frequency *f_o_* ≈ 23.33 Hz depending on the spindle rotational speed, then *f_oz_* ≈ 140 Hz frequency of the milling process and their harmonic values, these werethe basic signals depending on the kinematics of the process milling.

As presented in the research [21], this method can be successfully used in the detection of chatters, which in particular has a large impact when machining thin-walled workpieces.

In order to broaden the possibilities of predicting the deflection of thin walls during milling, the finite element method (FEM) implemented in the Fusion 360 program was used. Two attempts of the deflection analysis using this method were carried out. In the first (a), the wall was loaded with a constant force of 100 N obtained on the basis of measurements carried out in previous tests with machining with identical cutting conditions. The second attempt (b) consisted in applying pressure to the wall with a modeled tool (end mill) equal to the thickness of the cutting layer. This method more accurately represents the actual milling process in which, at a small thickness of the cutting layer (*a_e_*), the contact of the cutter with the machined wall is at the points where the cutter blades are in contact at a given moment. Figure 9 shows the simulation of both trials.

The values of deflections obtained on the basis of the simulation allowed to obtain two courses presented below (Figure 10 and Figure 11). Both tests a and b did not allow to predict the deflection of the tested wall in the full range of its thickness. The continuous load showed the deflection compliance within the range of 1 mm of the wall thickness, while the action of the cutter on the wall correctly showed the deflection for thickness values t above 1 mm.

A number of studies using the finite element method to predict thin wall deflections focus on one variant of a workpiece with a constant thickness, where the selection of the load method in the simulation is simplified. The variable deflection characteristics of the wall at different thicknesses make it difficult to predict deflection using the presented methods in this study.

## 4. Discussion

The aim of the research was to verify the usefulness of a high-speed camera in recording deformations of a thin-walled workpiece during peripheral milling. On the basis of the presented results, a difference of 11–28% can be noticed compared to the displacement laser sensor as the commonly used method for measuring thin elements. The error of 28% is when the deformation for a wall with a thickness of 0.62 mm is taken into account, for which the deformation value differs from the entire obtained plot, repeating the tests in a larger number of samples will result in obtaining greater accuracy, eliminating this error, at the same time, the calculation of the uncertainty of measurement is minor (Figure 12).

In addition, the FEM simulation in both variants allows to estimate the deformation in a way that allows for additional verification of the results obtained from the high-speed camera.

The method of applying forces to a thin wall is a difficult topic and is carried out by many researchers in a different way. The method of applying the forces depends on the type of the selected tool and the machining method (climb or up milling). Hence, it is planned to conduct tests using a two-edge end mill with the tool helix angle *λ_s_* = 0^o^ to facilitate the FEM simulation.

The authors [6] obtained the prediction of deformations with a inconsiderable error, but the way in which the force was a applied to the treated surfaces has not been specified, which would certainly serve and facilitate further research.

On the other hand, the tests [5], as described earlier, carried out a simulation in a manner close to the actual cutting conditions, allowing for tearing off a part of the material removed from the thin-wall, achieving high accuracy in predicting the deformation. Figure 11 shows the results of the simulation in which the modeled cutter plunges into the machined wall, unbending it. The obtained deformation values are overestimated by 50% in relation to the results obtained from the laser displacement sensor. This method has proven successful in the range of wall thicknesses below 1 mm, which at the thickness-to-height ratio is problematic in prediction due to the change in the dynamic characteristics of the wall.

The advantage of using a high-speed camera over a displacement laser sensor is the larger spectrum of data that can be obtained from this method. A high-speed camera allows to record a recording of the cutting process, which we can freely analyze with the appropriate software, recording the values of deformations, deflections, perform a frequency analysis, analysis of chip formation. Additionally, analysis of cutting force, torque and the amount of material removed. All these elements can make it possible to minimize the measuring equipment necessary to conduct research on the cutting process and machining of thin-walled workpieces.

Additionally, from the obtained registration of the course, we are able to determine the occurrence of plastic deformation of the workpiece without the need to perform meteorological measurements.

The use of a high-speed camera requires certain test conditions, such as: ensuring adequate lighting, free space in the camera’s working space, quality and type of the tested surface. The selection of the resolution and the number of frames per second with which the recording will be carried out affects the length of file processing by the software and their weight, which should also be taken into account when conducting tests. The need to use an additional light source had a negative impact on the measurements carried out by making it difficult to study the deformation of a very strongly illuminated wall with a laser sensor. In addition, during machining, the cutting chips fly into the space between the tested wall and the high-speed camera, causing the loss of the ability to track the marked point (upper corner of the wall), which resulted in the need to manually re-set the measurement point.

At the present moment, on the basis of the conducted research and the analyzed literature, it is difficult to identify the factors that have the greatest impact on the accuracy of measurements carried out with the use of high-speed cameras. On the basis of own research, a significant influence of the preparation of the side surface of the wall on the accuracy of the traceability of the point defined on it by the software was found. During further research, various methods of eliminating this problem will be tested by obtaining a matte surface with the use of matte paint and improving the quality of the surface by selecting an appropriate finishing machining when preparing the test sample.

The thin walls being the subject of the study were made on a sample having a rectangular base with smooth sides. As the research shows [11], the differences in the mounting of a very similar sample with thin walls influenced in less than 5% the difference in the frequency analysis, which allows for assuming no significant influence of the mounting method on the results obtained in the study.

The tests were carried out on a conventional AVIA FND 32 tool milling machine, the maximum rotational speed of the spindle of which didnot allow achieving the recommended cutting speed *v_c_*= 550 m/min for the selected cutter. However, the value of the feed per tooth was kept to maintain the recommended cutting conditions. These differences could affect the result of the deflection value, therefore further tests will be carried out on a milling center with a high-speed spindle. The choice of the machine tool was dictated by the large access to the working space of the tool, as shown in Figure 3, space is required in a certain range parallel to the processed wall, allowing for the sharpness of the image for the camera. In addition, a large part of numerically controlled machine tools has a sliding table in one or two axes, causing that the camera would have to move with the table or be on it in order not to lose the sharpness of the image.

Further research will focus on verifying the accuracy of thin-walled workpiece deformation measurements using high-speed cameras in both aluminum and titanium alloy machining. A more accurate FEM simulation method will be developed to predict the deformation of thin walls in a wide range of their dimensions. The possibility of measuring the deformation of thin elements in other places than from the front of the wall (for example closed thin-walled pockets) will be tested. All this will allow for the development of guidelines and instructions for the machining of thin-walled workpieces.

## 5. Conclusions

Based on the research, the following conclusions can be drawn:the high-speed camera can be successfully used to test thin wall deflections during milling, which is confirmed by the results compared to the measurement of displacements with a laser sensor with an accuracy of up to 11% and simulation by the finite element method with error of 22%,notwithstanding, it should be mentioned that the measurement itself requires specific conditions which include many factors that may introduce an error into the measurement or make it impossible,research confirms the dependence of nonlinearities in the deformation of thin walls under the influence of cutting forces, deformations below 1 mm of the wall thickness increase rapidly, which was noticed in previous studies, despite a different workpiece material such as hardened steel [28],the simulation carried out with the use of the finite element method, depending on the sample, made it possible to predict the deflection of thin-walled workpieces in a certain range of wall thickness,taking into account both simulation tests with the finite element method, the results obtained with them separately in the range of up to 1 mm of the wall thickness and below are estimated with an error of 22%,examination of deflections of thin-walled workpieces with a high-speed camera allows us to conduct a spectral analysis of their displacement, which makes it possible to diagnose them to an even greater extent, for example, for detecting chatter.

## Figures and Tables

**Figure 1 materials-14-04771-f001:**
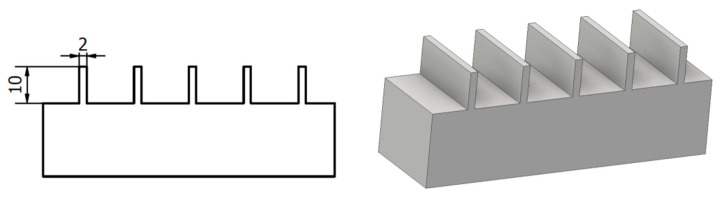
Research sample containing thin walls.

**Figure 2 materials-14-04771-f002:**
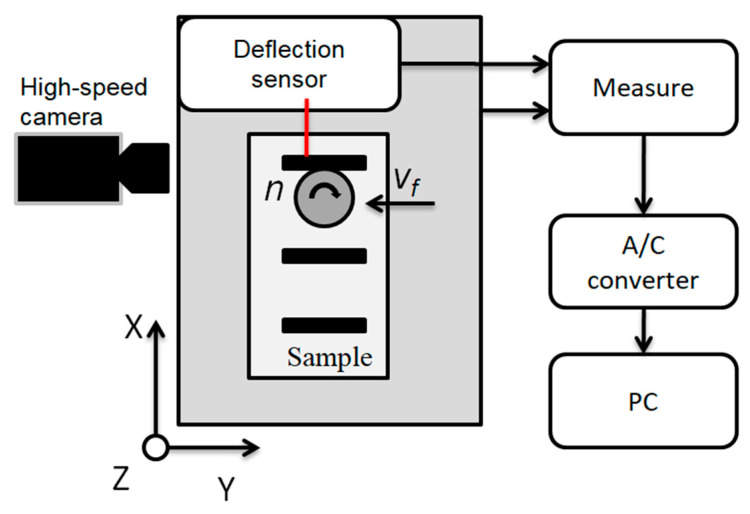
Experimental setup.

**Figure 3 materials-14-04771-f003:**
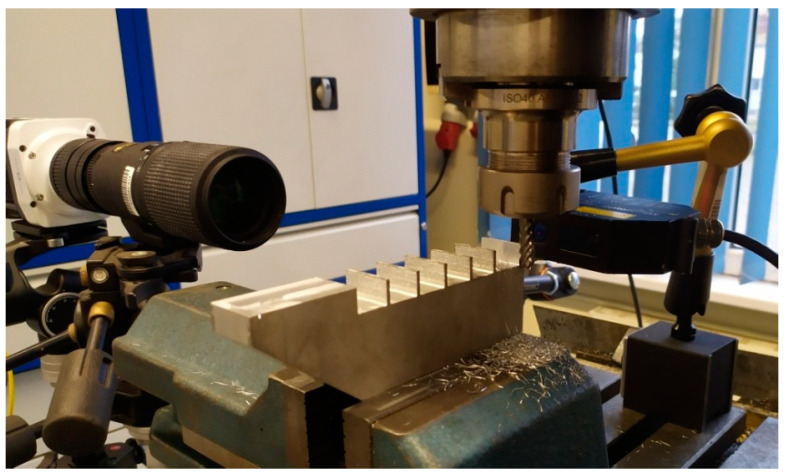
Photo of test stand.

**Figure 4 materials-14-04771-f004:**
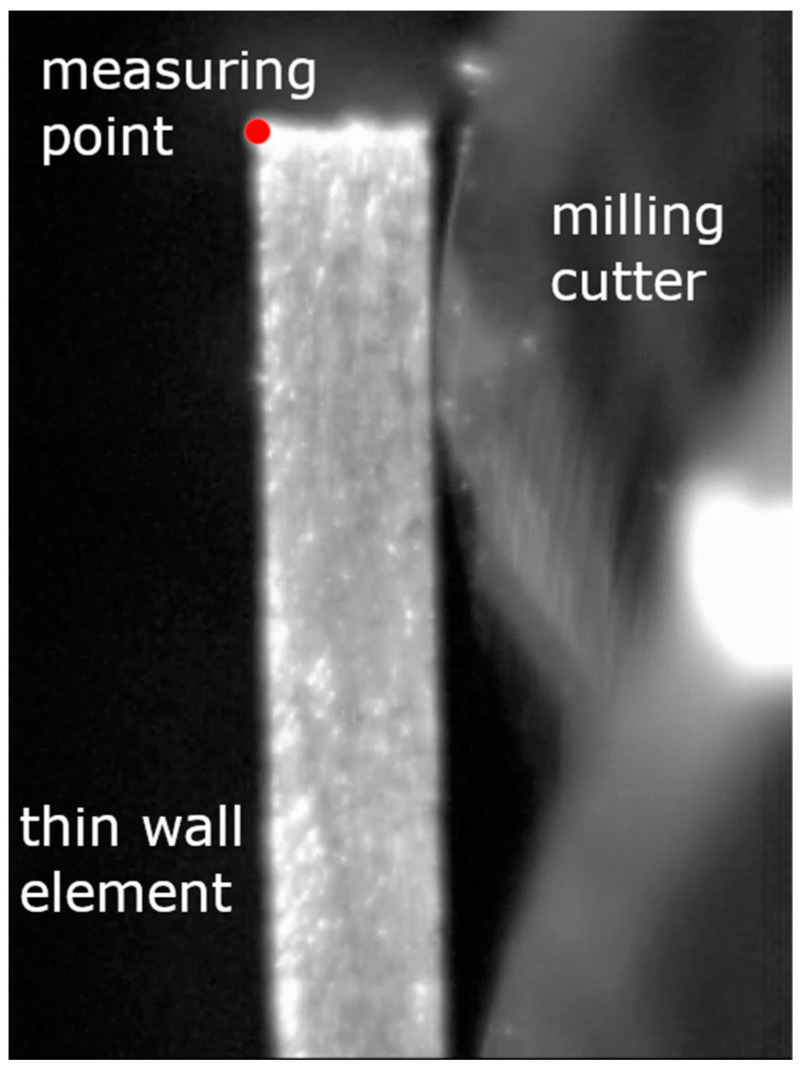
A frame of film recorded with a high-speed camera while milling a thin wall.

**Figure 5 materials-14-04771-f005:**
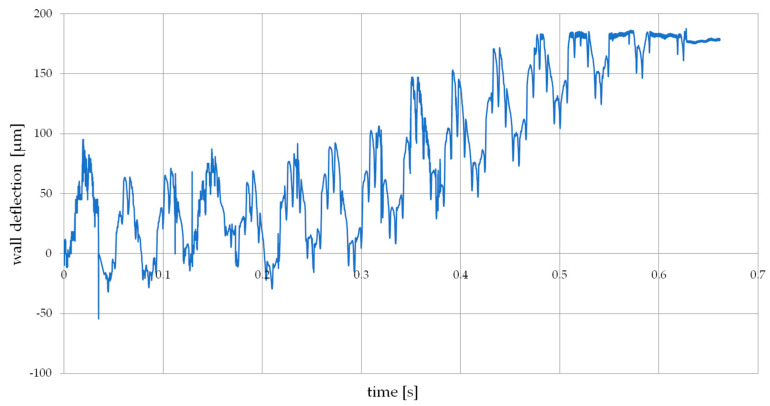
The plot of the deflection of the thin-walled workpiece obtained on the basis of the recording analysis.

**Figure 6 materials-14-04771-f006:**
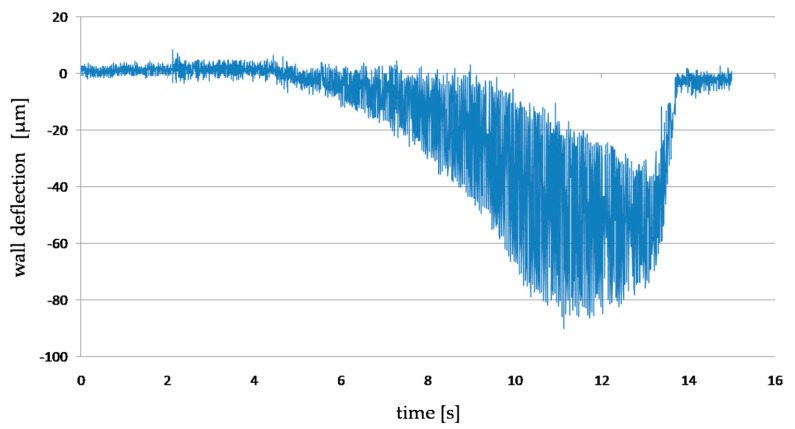
The plot of the deflection of the thin-walled workpiece obtained by measuring the deformation with a displacement laser sensor.

**Figure 7 materials-14-04771-f007:**
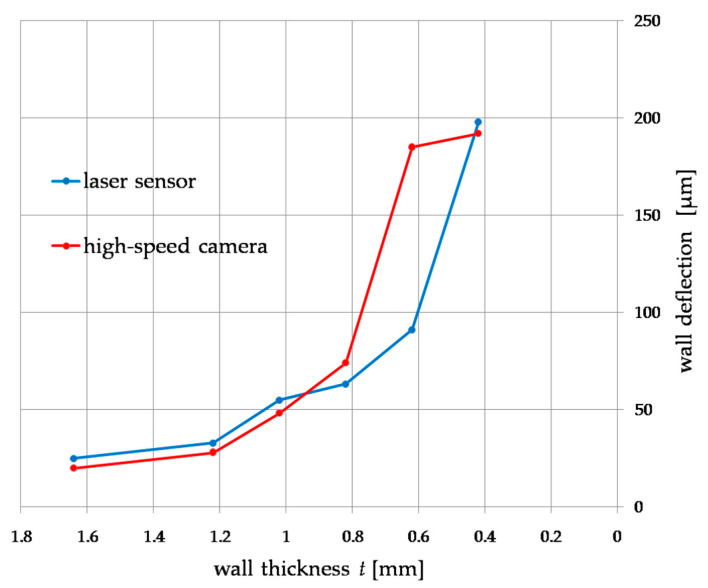
Comparison of the results of the measurement of deflection during thin-wall milling measured with a laser displacement sensor and a high-speed camera.

**Figure 8 materials-14-04771-f008:**
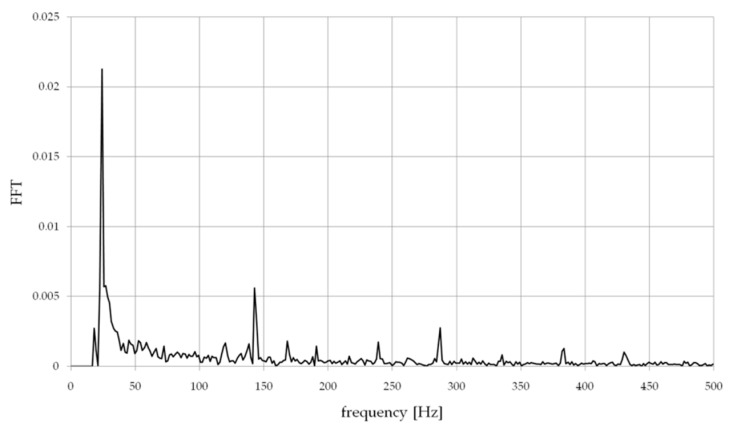
An example of the amplitude-frequency analysis of the displacement of the milled wall.

**Figure 9 materials-14-04771-f009:**
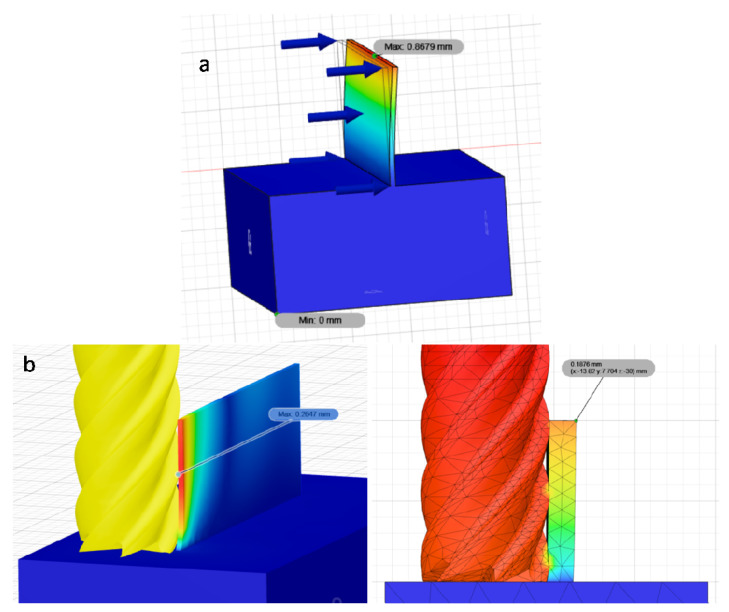
Thin wall deflection simulation made in Fusion 360: (**a**) constant force 100 N; (**b**) pressure of the end mill on the thin wall.

**Figure 10 materials-14-04771-f010:**
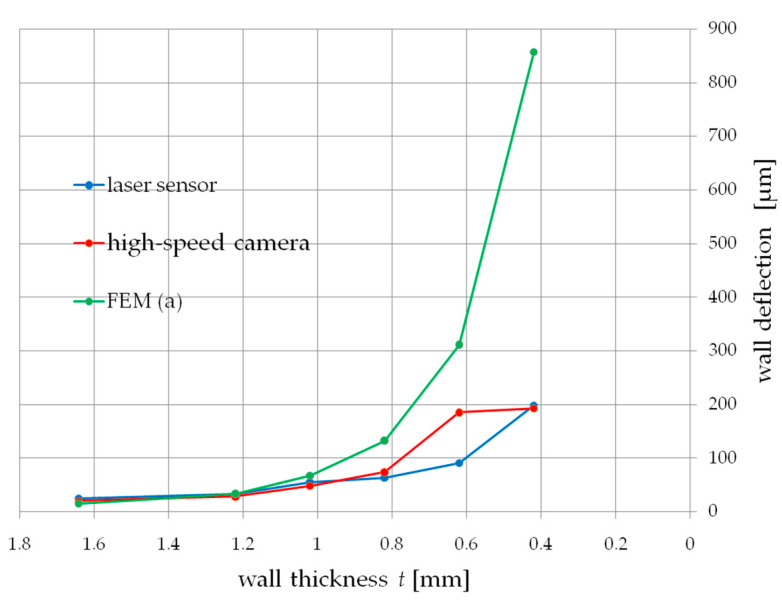
Finite element analysis—wall deflection under the influence of a constant force, test a.

**Figure 11 materials-14-04771-f011:**
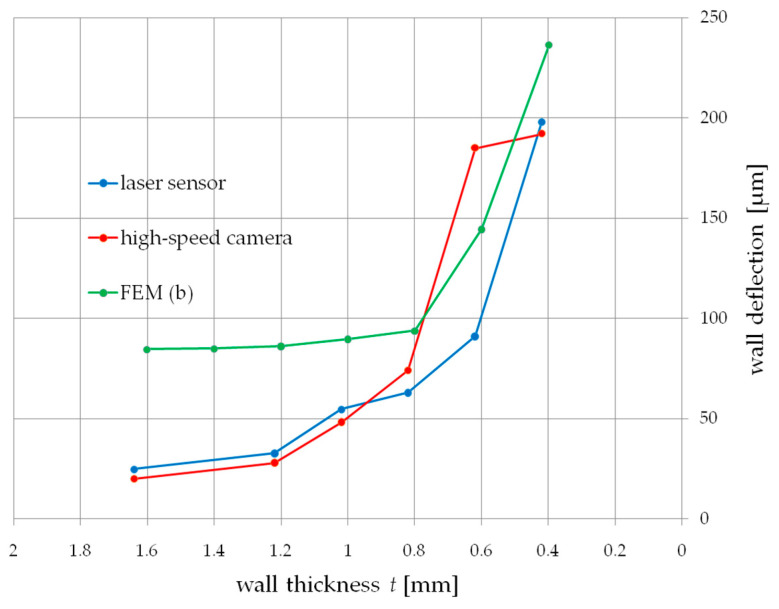
Finite element analysis—wall deflection under influence of the cutter, test b.

**Figure 12 materials-14-04771-f012:**
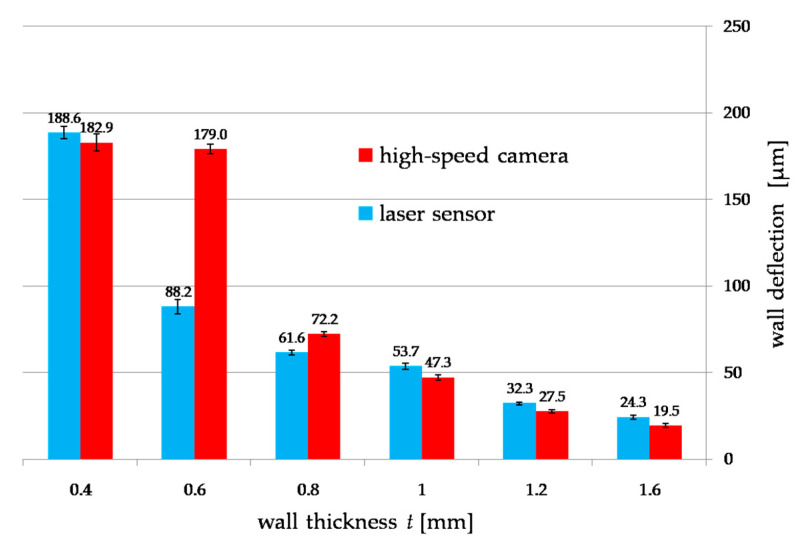
Measurement uncertainty when measuring thin walled workpiece deflection with a high-speed camera and displacement laser sensor.

## Data Availability

The data presented in this study are available on request from the corresponding author.

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
