# Peer review of "Analysis of the Displacement of Thin-Walled Workpiece Using a High-Speed Camera during Peripheral Milling of Aluminum Alloys"

_materials, 2021, doi:10.3390/ma14164771_

Round 1

Reviewer 1 Report

In this study, the effectiveness of a high-speed camera to analyze the displacement of thin-walled elements during milling was demonstrated well, so the manuscript is considered to be worth publishing in this journal. Before acceptance, several points listed below should be addressed.

  1. The unit of the vertical axis in Fig.5 should be “μm”, not “mm”, I think. Please check it.
  2. For the future work, the reasons for discrepancy of results between FEM (a)/(b) and the measured values should be discussed in detail, and possible ideas to overcome these discrepancies should be proposed, if possible.

Author Response

Thank you for your careful comments, the paper has been improved.

Reviewer 2 Report

The article must be significantly supplemented and corrected. Some of the suggestions for improvement are as follows:

  1. The title of the article should be corrected. You do not analyse all types of milling.
  2. The abstract should clearly state the essence of the problem you are addressing, what you did and what you found and recommend.
  3. What is the scientific contribution? What are the scientific hypotheses of this research?
  4. The section pertaining to review of previous contributions in the field must be expanded with other journal references.
  5. It is necessary to present in detail the scientific contributions and results of previous research. Then do a critical analysis of previous research. State explicitly the shortcomings of previous research.
  6. The Materials and Methods section needs to be significantly adjusted to see the universality of the methodology applied.

The choice of technological parameters must be explained in detail.

The initial wall thickness t was 2 mm. Explain the choice.

The wall height and hence the cutting depth ap =10 mm. Explain the choice.

The cutting width being the thickness of the layer of the cut material during a single pass was ae=0.2 mm. Explain the choice

The machining was carried out at the cutting speed vc = 110 m/min for feed per tooth fz=0.13 mm/tooth. Explain the choice

Why are these representatives’ technological parameters? Elaborate further.

Also, further elaborate the choice of end mill parameters (diameter, number of teeth, tool helix angle)

  1. “Thin-walled element” or “Thin-walled workpiece”?
  2. In the Results and discussion section, the authors only discuss the results descriptively. That's not enough. The research must have a special section Discussions in which the results are SCIENTIFICALLY DISCUSSED and compared with previous research. The conducted research should be critically analyzed. The authors should not only point out the advantages but also the disadvantages.
  3. The biggest drawback of this research is that there is no analysis of potential errors.
  4. The authors would have to calculate the measurement uncertainty. Without the calculated measurement uncertainty, the obtained results are not complete.
  5. Will the change of measuring instrumentation give identical or similar results? This is very important to discuss further due to the universality of the methodology.
  6. Discuss the impact of fixture on the results obtained. Pay special attention to changing the points of the location elements and clamping elements.
  7. The conclusions should indicate the limitations of the applied methodology and potential future research. Also, the authors should list the benefits of this research. Benefits must be proven and not just declaratively written.

Author Response

Thank you for the nice words. In a final form the paper has been improved according to the reviewer remarks.

Reviewer 3 Report

The present article deals with the monitoring of the process of machining thin walls of aluminum alloy products. This process hides many specific properties and events, so it is necessary to address the issue. The authors appropriately processed the issue of monitoring using a thermal camera, they devoted themselves to a wide literature search, which significantly contributed to the quality of the article. I have no scientific or formal comments  and I recommend publishing.

Author Response

Thank you for the nice words. Research will be developed more and more in order to improve the accuracy of prediction of thin-walled elements deformation. 

Reviewer 4 Report

The results are interesting. The present submission reports the analysis of the displacement of thin-walled aluminum alloys. By utilizing a high-speed camera during milling, the investigators were able to compare to other methods of testing the deflection. During the experiment, the authors further pointed out some other pieces of information that may help other researchers, e.g. additional light sources and appropriate distance. However, the physics or the science behind the experiments needs to be clarified with the interpretation of the results. Therefore, I do not feel this paper is ready for publication without revision and corrections.

  1. Technically, it is a good piece of work. The major deficiency in this manuscript is that this study is not yet generalized. The reviewer understands the possible advantages of the high-speed camera over other methods which is the major point of this study. Using 7075 as a model system is a good choice as well. But solely the fact that the high-speed camera can be used for analysis of the displacement is not enough to warrant the publication. More thorough interpretation is needed to make this manuscript better. Interesting discussions are - what is the pros/cons of the method compared to other methods measuring displacement, what are the other data that can be explicitly obtained from this method, or (if the authors want to discuss on materials point of view) how the deflection may vary among the materials, or what is the relationship between property vs deflection, etc.
  2. (Fig. 5) At 0.55 s (and over), the wall deflection gives ~170 and does not change over time. Here, is the deformation in the manuscript “elastic deformation”? Since the number is ~ 170 over 0.55 s, does the "170" imply plastic deformation? The reviewer just wants to clarify whether understood correctly. From the manuscript, “The following test results show the maximum deflection of a thin wall at its top at the stage of completing a single end mill pass by cutting a given material thickness. This value is of great importance because it allows determining whether the deformation is large enough to reduce the assumed cutting layer (ae), which will cause an error in the thickness of the wall being made, as proven in the tests”. To make it clear, it may be better point out which “deformation” it is – elastic or plastic.   
  3. (Fig. 6) How large the error will it be from the measurement? The reviewer believes that the authors might use the system multiple times to measure/set up the deflection. It would be helpful to show the measured error ranges of data obtained from the camera and error bars in the graphs.
  4. (page 4)“increase in strains of up to 200 um”. “STRAIN” in science is the ratio of total deformation to the initial dimension of the material body on which forces are applied.
  5. Please double-check the typos- for example, in fig 5. “wall deflection [mm]”.

Author Response

(The authors gave the same response as above.)

Reviewer 5 Report

The article is an interesting aproach on the possibilities of a high-speed camera in recording displacements of thin-walled elements during milling made of aluminum alloys, which allowed for an analysis in which it was compared to other methods of testing the deflection of such elements.

The article is well presented and the research is sound. 

The literature should be more detailed and presented more researches from last 4 years. 

My only doubt is if Materials Journal is suitable, or a different like Applied Sciences and on mechanical issues. 

Author Response

Thank you for the nice words. In a final form the paper has been improved according to the reviewer remarks.

Four recently published articles were added as recommended, three of which have been described. The use of HSM (high-speed machining) in the machining of thin-walled elements was clarified, the method of detecting chatter possible for industrial applications was described, and the costs associated with the purchase of a high-speed camera and the possibility of making such a camera at a fraction of the price of cameras available on the market were described. 

The current state of literature does not present articles focusing on the study of deformation of thin-walled elements during their cutting with the use of a high-speed camera. The article presents the use of high-speed cameras in machining and its intermediate areas. There is, however, a significant amount of research on thin-walled elements, from which the ones considered important for the topic of our research were selected.

If there are articles worth mentioning, please indicate them. 

Article will be published in the Special Issue: "Innovative and Modern Technologies of Material Machining in Cutting and Abrasive Processes"

Round 2

Reviewer 1 Report

The revised manuscript is now considered to be acceptable for publication.

Reviewer 2 Report

The manuscript has been updated.